# Genetic Differences among Established Populations of *Aromia bungii* (Faldermann, 1835) (Coleoptera: Cerambycidae) in Japan: Suggestion of Multiple Introductions

**DOI:** 10.3390/insects13020217

**Published:** 2022-02-21

**Authors:** Shigeaki Tamura, Etsuko Shoda-Kagaya

**Affiliations:** Department of Forest Entomology, Forestry and Forest Products Research Institute, Forest Research and Management Organization, Matsunosato 1, Tsukuba 305-8687, Japan; eteshoda@affrc.go.jp

**Keywords:** genetic analyses, invasive pests, mitochondrial DNA, red-necked longicorn beetle, Rosaceae, wood-boring insect

## Abstract

**Simple Summary:**

The red-necked longhorn beetle, *Aromia bungii*, is a pest that damages Rosaceae trees such as peach and cherry. This beetle has been introduced into Japan and expanded its distribution rapidly during the last decade. Currently, Japanese populations of *A. bungii* are widely distributed in six non-contiguous regions. The DNA sequences of partial mitochondrial DNA were analysed in Japanese populations of *A. bungii* in order to discuss whether multiple introductions or human-mediated long-distance dispersal have contributed to the non-contiguous distribution of *A. bungii*. Seven haplotypes were detected in Japanese populations. Haplotype composition differed among the six non-contiguous regions. These results suggest that multiple introductions have contributed to the non-contiguous distribution of *A. bungii* in Japanese populations.

**Abstract:**

*Aromia bungii* (Faldermann) (Coleoptera: Cerambycidae) is an invasive pest, damaging Rosaceae trees (particularly *Prunus*) in Japan and Europe. The establishment of this beetle in Japan was first detected in 2012, and subsequently, it has rapidly expanded its distribution. Currently, Japanese populations of *A. bungii* are widely distributed in six non-contiguous regions. In this study, we compared the nucleotide sequences of mitochondrial cytochrome oxidase subunit 1 of the populations in these six regions in Japan to examine whether multiple introductions or human-mediated long-distance dispersal have contributed to the non-contiguous distribution of *A. bungii*. Seven haplotypes were detected from Japanese populations, and one of these was identical to a sequence deposited from China. One to two haplotypes were detected in each region, suggesting a genetic bottleneck. Detected haplotypes differed between introduced regions, although two regions shared a single haplotype. These results suggest that multiple independent introductions of *A. bungii* have contributed to its non-contiguous distribution in Japan. Quarantine measures for wood-packing materials in trade need to be strengthened to prevent the establishment of further populations of *A. bungii*.

## 1. Introduction

The red-necked longicorn beetle, *Aromia bungii* (Faldermann, 1835) (Coleoptera: Cerambycidae), is an invasive wood-boring pest, which has been recently introduced into Germany, Italy, and Japan from countries within its original range, such as China, Korea, Mongolia, and Vietnam [1]. The hosts of *A. bungii* are primarily Rosaceae trees, particularly *Prunus* species [1], and the larvae of *A. bungii* often severely damage host trees by feeding on the inner bark and cambium, and by boring into the xylem of the trees. Therefore, they cause economic losses by decreasing the yields of Rosaceae fruit crops, such as peaches and apricots, both in their countries of origin and where they have been introduced [1,2,3]. Additionally, ecosystem disturbance by *A. bungii* is a concern in Japan because *A. bungii* can grow in the logs of Rosaceae trees in natural forests [4].

In Japan, the establishment of *A. bungii* was first detected in 2012 [5]. By 2019, *A. bungii* was widely distributed in six non-contiguous regions of Japan: northern Kanto, southern Saitama, western Tokyo, southern Tokai, western Kansai, and northern Tokushima [6,7] (Figure 1). The establishment of this beetle was first detected in southern Tokai in 2012, southern Saitama in 2013, and the other four regions in 2015 [7]. In all of these regions, *A. bungii* was first detected because of damage caused to ornamental cherry trees (e.g., *Cerasus × yedoensis* ‘Somei-yoshino’) planted in urban areas [5,8,9,10]. In northern Tokushima, peach (*Prunus persica*) and Japanese apricot (*P. mume*) trees in orchards were also damaged by *A. bungii* [11]. Distributions of *A. bungii* have gradually expanded in all introduced regions [7].

Exotic wood-boring pests are often introduced in wood-packing materials, and their invasion has increased with increasing trade [12]. Recently, multiple introductions of exotic wood-boring insects have been reported, which are thought to have been introduced in wood-packing materials [13,14,15,16,17]. Multiple introductions may have contributed to the extremely wide distribution of the Asian long-horned beetle (*Anoplophora glabripennis*) in Europe and North America [15,16]. Additionally, *A. bungii* may have been introduced via the presence of its larval and pupal stages in wood-packing materials [1]. For example, individuals of *A. bungii* have been intercepted on wood-packing materials in Washington State in the United States of America, Baden-Württemberg in Germany, and in the United Kingdom [18]. The non-contiguous distribution of *A. bungii* in Japan suggests multiple independent introductions in each region. However, Iwata [6] pointed out the possibility that the non-contiguous distribution of *A. bungii* was formed by human-mediated long-distance dispersal, caused by the species ‘hitchhiking’ on vehicles. Understanding whether multiple introductions or human-mediated long-distance dispersal has contributed to the distribution of *A. bungii* could be useful for the management of this beetle.

In this study, we compared nucleotide sequences of the mitochondrial cytochrome oxidase subunit 1 (CO1) region of *A. bungii* populations in Japan to examine whether multiple introductions and establishments or human-mediated long-distance dispersal has contributed to the non-contiguous distribution of *A. bungii*. If multiple independent introductions of *A. bungii* occurred in Japan, then there would be genetic differences between invaded regions. However, if there was long-distance dispersal between regions, then populations would be genetically similar between invaded regions.

## 2. Materials and Methods

We collected 4–45 individuals of *A. bungii* from 1 to 14 sites in all six introduced regions in 2015–2019, resulting in a total of 120 individuals (101 adults, 18 larvae, and 1 pupa) being collected from 37 sites in Japan (Table 1, Appendix A, Figure 2). Of the 101 adults, 85 were collected in the field and 16 emerged from the logs of damaged trees transported to the laboratory. All larvae and the single pupa were collected by cutting trees or peeling bark. The adults collected in northern Kanto, southern Saitama, western Tokyo, and northern Tokushima were frozen immediately after transporting them to our laboratory. The adults collected in Southern Tokai and western Kansai were fixed in 99.5% ethanol or 98% propylene glycol immediately after collection. All larvae and the single pupa were fixed in 99.5% ethanol immediately after bringing them to our laboratory. We preserved all samples at −30 °C in freezers until DNA extraction. We did not notice a difference in the morphology of collected adults, such as body size and coloration, among invaded regions.

We sequenced two sections of the mitochondrial DNA CO1 regions (612 and 762 bp) of all collected individuals. In our preliminary analysis, we found only a few nucleotide substitutions in the 612-bp section. Thus, we additionally analyzed the 762-bp section. We used PrepMan Ultra Sample Preparation Reagent (Applied Biosystems, Foster City, CA, USA) to extract DNA from muscle tissues. Dried tissues were added to 75 µL PrepMan liquid in 1.5 mL tubes and then heated at 100 °C for 10 min. Then, we centrifuged the tubes at 14,000 rpm (17,800× *g*) for 3 min via a centrifuge (mx-301, Tomy Seiko, Japan), and diluted the supernatant liquid tenfold with MilliQ water. MilliQ water was produced by a Synergy UV Water Purification System (Millipore, Burlington, MA, USA). We used the diluted liquid as extracted DNA for polymerase chain reaction (PCR) without measuring the quality of extracted DNA by spectrophotometer.

We mixed PCR reagents for each reaction: 0.13 µL of TaKaRa Ex Taq (TaKaRa Bio, Japan) or TaKaRa Ex Taq Hot Start ver. (TaKaRa Bio, Japan); 2 µL of 10 × Ex Taq Buffer; 1.6 µL of dNTP mixture; 2 µL of each primer (100 µM/L); and 13 µL of MilliQ water. We used two primer pairs: LCO1490 (5′-GGTCAACAAATCATAAAGATATTGG-3′) and HCO2198 (5′-TAAACTTCAGGGTGACCAAAAAATCA-3′) [19] for the shorter section, and CO1-Croz (5′-CAACATTTATTTTGATTTTTTGGTCA-3) [20] and tRNA^Leu^-R (5′-GGGGTTTAAATCCATTGCAC-3′) for the longer section. The primer tRNA^Leu^-R was the reverse sequence of tRNA^Leu^-F, and was designed by Kawai et al. [21]. We added 17 µL of mixed PCR regents and 0.8 µL of extracted DNA to 0.2 mL tubes. We performed PCR amplification of the profile as follows: 30 cycles of 94 °C for 30 s, 45 °C for 30 s, and 72 °C for 110 s, with initial denaturation at 94 °C for 3 min and a final extension at 72 °C for 7 min. We confirmed the success of amplification using electrophoresis into 1.5% agarose gel with Tris-Borate-EDTA buffer and Invitrogen SYBR Safe DNA gel stain (Thermo Fisher Scientific, Waltham, MA, USA). We re-extracted the DNA of any samples for which amplification failed (17 individuals) using the DNeasy Blood and Tissue Kit (Qiagen, Valencia, CA, USA) and performed PCR under the same conditions. Hence, the success rates of PCR amplification were 86% (103/120) in the DNA extracts by PrepMan and 100% (17/17) in the DNA extracts by DNeasy in this study.

We purified PCR products by ethanol precipitation coupled with polyethylene glycol precipitation, or using ExoSAP-IT Express PCR Cleanup Reagents (Applied Biosystems, Foster City, CA, USA). In the precipitation method, we added 13% PEG8000/1.6 M NaCl solution into PCR products at an equal ratio. After resting the products on ice for longer than 1 h, we centrifuged them at 14,000 rpm (17,800× *g*) and 4 °C for 60 min via the centrifuge. We removed the supernatant liquid and added 75 µL of 70% ethanol. Then, we performed centrifugation at 14,000 rpm (17,800× *g*) and 4 °C for 30 min via the centrifuge. We removed the supernatant liquid and dried the precipitated DNA fragments. Then, we dissolved the DNA fragments in 5–10 µL MilliQ water and sequenced the purified DNA fragments using a BigDye Terminator v3.1 Cycle Sequencing Kit (Applied Biosystems, Foster City, CA, USA) and ABI 3130 Genetic Analyzer (Applied Biosystems, Foster City, CAA, USA). We aligned the obtained nucleotide sequences using BioEdit version 7.2.5 software [22]. The obtained mitochondrial CO1 sequences were deposited in the DNA Data Bank of Japan, European Molecular Biology Laboratory, and GenBank (accession nos. LC574081–LC574320).

We compared the haplotype compositions of populations of *A. bungii* to examine the genetic differences between invaded regions in Japan. We also tested the genetic differences between populations of northern Kanto and southern Saitama where we collected many samples, using an analysis of molecular variance (AMOVA) for mitochondrial CO1 sequences based on pairwise nucleotide differences. We partitioned the total genetic variance into that between regions, among sites within regions, and within sites. Then, we evaluated the molecular variance among groups (Φ_CT_) and within groups (Φ_SC_), based on permutations among 10,000 replications.

We compared the sequences of Japanese individuals of *A. bungii* with those deposited in GenBank from the countries of origin to examine the relationship between the Japanese populations and populations from the countries of origin. We selected the following deposited sequences that overlapped with the short and long sequences of this study in more than 1000 bp in total; accession nos. MT371041 and KF737790 from China, and OK428926–OK429124 from Korea. We constructed haplotype networks based on 1170-bp sequences of mitochondrial CO1 shared between the sequences of this study and previously deposited ones. We used the *pegas* package [23] for R version 4.0.0 software [24] to construct the haplotype network and to perform the AMOVA.

## 3. Results

Seven haplotypes, designated A–G, were detected in analysed Japanese individuals of *A. bungii*, and there were fewer than 0.7% (10 bp) differences in bases among these haplotypes (Table 2). Haplotype B, which was found in northern Kanto, was consistent with a sequence deposited from China (accession no. KF737790) (Figure 3). Other deposited sequences from China and Korea (MT371041 and OK428926–OK429124) were not consistent with the sequences found in this study.

Only a single haplotype was detected in populations from southern Saitama (haplotype C), western Tokyo (haplotype A), southern Tokai (haplotype D), and northern Tokushima (haplotype G) (Table 1, Figure 1). Multiple haplotypes were found in populations from northern Kanto (haplotypes A and B) and western Kansai (E and F). Two haplotypes often coexisted within populations in these regions (Table 1). The detected haplotypes were different between invaded regions, with the exception of the populations in northern Kanto and western Tokyo, which shared haplotype A. The populations of *A. bungii* were genetically different between northern Kanto and southern Saitama (Table 3). The genetic variance of Japanese *A. bungii* primarily resulted from variance between regions (72%), and there was low genetic difference among sites within regions (3%).

## 4. Discussion

We detected seven haplotypes from populations of *A. bungii* in Japan using mitochondrial CO1 sequence analyses, and one to two haplotypes were found in each region. Similar to many alien species, this low diversity in CO1 sequences suggests a bottleneck in the Japanese populations of *A. bungii* [25]. Populations of *A. bungii* could have been established by a small number of females in each region.

All the haplotypes found in Japan were not consistent with the sequences deposited from South Korea, even though the diversity of CO1 sequences of *A. bungii* has been investigated in detail in South Korea [26]. The CO1 sequence of haplotype B, which was found in Japan, was identical to the sequence deposited from China. These results suggest that Japanese populations of *A. bungii* have not originated from South Korea, and that at the very least, the populations in the northern Kanto region were introduced from China. However, detailed analyses of the genetic structure of populations in China, Mongolia, and Vietnam are needed to identify the area of origin of Japanese populations of *A. bungii*.

The detected haplotypes differed between the introduced regions, except for haplotype A, which was shared between northern Kanto and western Tokyo. Additionally, genetic differences were observed between northern Kanto and southern Saitama. These results strongly suggest that multiple introductions of *A. bungii* occurred in Japan. If populations of *A. bungii* independently established in southern Saitama, southern Tokai, western Kansai, northern Tokushima, and the other two regions, there could have been five establishment events.

The frequency of independent introductions over a period of less than 10 years in Japan could have resulted from the strong establishment ability of *A. bungii*, high propagule pressure, and the high susceptibility of Japanese urban habitats to this species. Females of *A. bungii* lay 300–500 eggs on average [27,28,29], which is high among cerambycid species [30]. Since males of *A. bungii* secrete sex-aggregation pheromones [31,32], mating opportunities may not be substantially decreased by a low population density. These traits of *A. bungii* suggest that they have a high ability to establish in new areas where they arrive [12,33]. The invasion pathway of *A. bungii* is thought to be via wood-packing materials used for trade [18], and the rate of imports to Japan from countries in the original *A. bungii* range, except for Mongolia, has recently increased (Figure 4). In particular, imports from China, where *A. bungii* is a common pest, have rapidly increased since 2000. Therefore, the propagule pressure of *A. bungii* has likely increased dramatically in the last two decades. Additionally, the high abundance of *A. bungii* host plants in Japan may also elevate Japan’s susceptibility to invasions. Many ornamental cherry trees are planted in urban areas throughout Japan because of the unique Japanese traditional culture of the spring cherry blossom festival (Hanami in Japanese) (Figure 5) [34,35]. Ornamental cherry trees have been damaged by *A. bungii* since the species was first detected in each of the introduced regions of Japan. Outside of Japan, in Italy, *A. bungii* distributes in the 250-km^2^ area of the Campania region, and only one haplotype has been detected in this area [29]. This supports a single introduction of *A. bungii* in Italy. In Germany, the distribution of *A. bungii* has been limited to a narrow area [1,36,37]. Compared to these two European countries, the frequent establishment of *A. bungii* in various locations is uniquely characteristic of Japan. It seems that the combination of high propagule pressure and abundant host trees may have encouraged the frequent establishment of *A. bungii* in Japan.

One haplotype was common between northern Kanto and western Tokyo, which are 30–50 km apart. This finding could be explained by long-distance dispersal mediated by humans, which has been reported for many exotic insects (e.g., [38,39,40]). Additionally, an adult of *A. bungii* was observed attached to a truck in Japan [41]. Therefore, it is feasible that *A. bungii* individuals have been accidentally transported between these introduced regions by vehicles. However, it remains probable that individuals of *A. bungii* have been independently introduced into these regions from populations in their original distribution range. Further studies using more detailed DNA markers (e.g., microsatellite markers) are needed to clarify this.

We found that the CO1 sequences of *A. bungii* are different between non-contiguous regions in Japan, suggesting that multiple introductions of *A. bungii* have contributed to the wide distribution of this beetle, less than a decade after the detection of its establishment. Since 2007, the Japanese government has required exporting countries to treat wood-packing materials in accordance with the International Standards for Phytosanitary Measure No. 15 (ISPM 15) [42]. However, the ISPM 15 treatment does not kill all wood-boring insects [43,44]. Therefore, quarantine measures for wood-packing materials and the early detection of new invasions of *A. bungii* around cargo destinations need to be strengthened to prevent further establishments of *A. bungii* into uninvaded regions in Japan.

## Figures and Tables

**Figure 1 insects-13-00217-f001:**
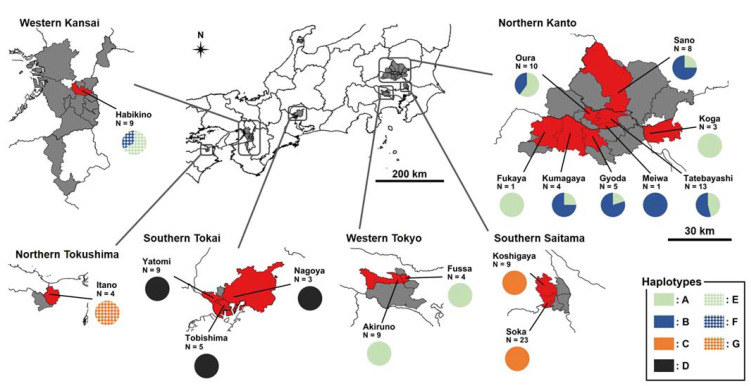
The distribution of *Aromia bungii* and its haplotypes based on mitochondrial CO1 sequences in Japan. Red shading indicates municipalities where individuals were sampled, and grey shading indicates municipalities where other populations of *A. bungii* were recorded as being established in 2019. Haplotype compositions of the studied municipalities have been combined for multiple study sites. N indicates the number of analysed individuals of *A. bungii*. All enlarged maps have the same scale. Maps were drawn based on those produced by the Digital National Land Information (administrative division data) and the Ministry of Land, Infrastructure, Transport, and Tourism Japan: https://nlftp.mlit.go.jp/ksj/index.html (accessed on 6 June 2021).

**Figure 2 insects-13-00217-f002:**
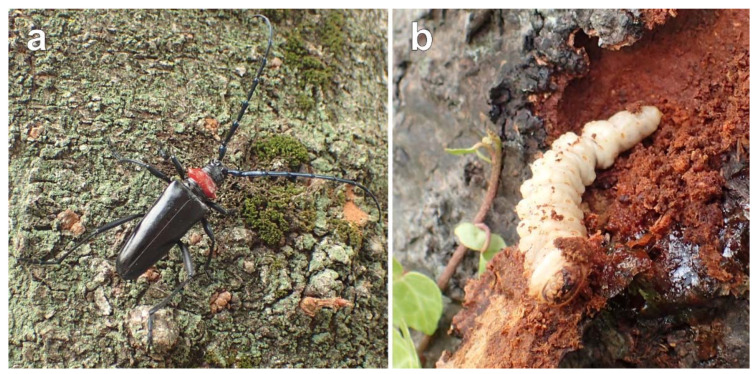
Photos of collected individuals of *Aromia bungii* in the field. (**a**) An adult female collected on the bark of a cherry tree in Tatebayashi City, Gunma Prefecture, on 28 June 2018. (**b**) A larva collected by peeling bark on a cherry tree in Fussa City, Tokyo Metropolis, on 22 May 2019.

**Figure 3 insects-13-00217-f003:**
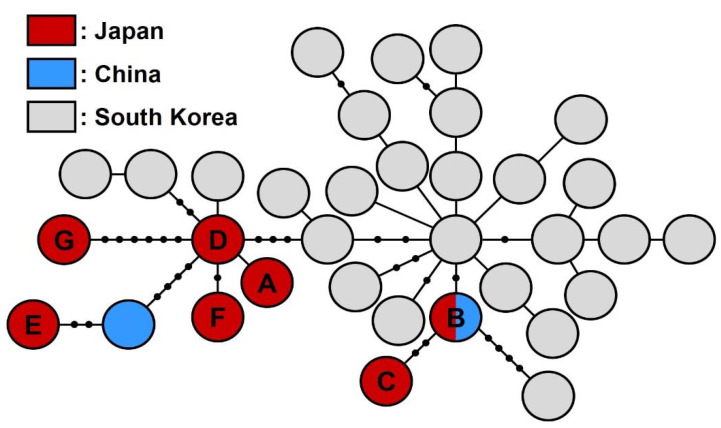
The haplotype network based on mitochondrial CO1 sequences (1170 bp) in Japanese, Chinese, and Korean *Aromia bungii*. Circles indicate haplotypes detected in this and previous studies. Black dots indicate potential haplotypes that were not collected. The seven haplotypes detected in this study were designated A–G.

**Figure 4 insects-13-00217-f004:**
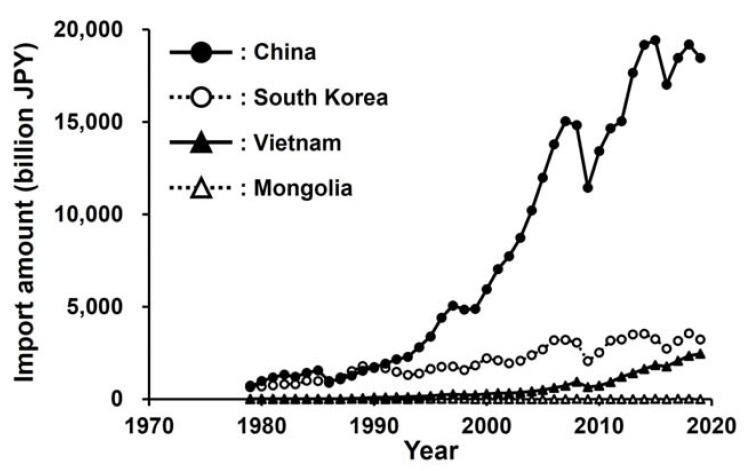
Import amounts from countries in the original range of *Aromia bungii* to Japan. This figure was drawn based on time series data obtained from Trade Statistics, Ministry of Finance Japan: https://www.customs.go.jp/toukei/suii/html/time_e.htm (accessed on 20 July 2020).

**Figure 5 insects-13-00217-f005:**
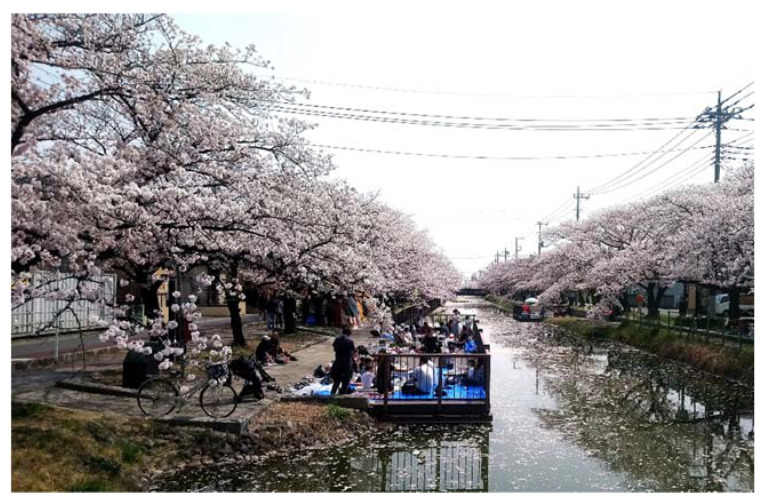
The cherry blossom festival along the Kasai Canal in Soka, Saitama Prefecture. Several hundred cherry trees are planted along the canal; people enjoy their time among the blossoms. This canal is the site where established *Aromia bungii* was first detected in southern Saitama in 2013.

**Table 1 insects-13-00217-t001:** Sampling information and haplotype compositions of *Aromia bungii*.

Introduced Region	Prefecture	Municipality	Site	Longitude	Latitude	Year	Haplotype Composition						
							A	B	C	D	E	F	G
Northern Kanto	Gunma	Tatebayashi	GTa1	36°14.6′ N	139°30.8′ E	2018	3	0	0	0	0	0	0
			GTa2	36°14.5′ N	139°32.8′ E	2018	1	1	0	0	0	0	0
			GTa3	36°13.7′ N	139°32.0′ E	2018	2	6	0	0	0	0	0
		Oura	GOu1	36°14.3′ N	139°29.5′ E	2019	3	2	0	0	0	0	0
			GOu2	36°15.2′ N	139°28.3′ E	2019	2	0	0	0	0	0	0
			GOu3	36°16.2′ N	139°26.9′ E	2019	1	2	0	0	0	0	0
		Meiwa	GMe1	36°12.5′ N	139°31.8′ E	2018	0	1	0	0	0	0	0
	Tochigi	Sano	TSa1	36°17.0′ N	139°32.8′ E	2018	0	1	0	0	0	0	0
						2019	2	5	0	0	0	0	0
	Ibaraki	Koga	IKo1	36°10.8′ N	139°42.1′ E	2019	3	0	0	0	0	0	0
	Saitama	Fukaya	SFu1	36°10.6′ N	139°13.6′ E	2019	1	0	0	0	0	0	0
		Kumagaya	SKu1	36°09.7′ N	139°24.7′ E	2019	1	3	0	0	0	0	0
		Gyoda	SGy1	36°10.9′ N	139°28.4′ E	2019	1	4	0	0	0	0	0
Southern Saitama	Saitama	Soka	SSo1	35°50.4′ N	139°47.7′ E	2017	0	0	2	0	0	0	0
			SSo2	35°50.2′ N	139°49.1′ E	2018	0	0	3	0	0	0	0
			SSo3	35°50.7′ N	139°49.7′ E	2018	0	0	2	0	0	0	0
			SSo4	35°50.8′ N	139°49.8′ E	2018	0	0	1	0	0	0	0
			SSo5	35°50.8′ N	139°49.5′ E	2019	0	0	3	0	0	0	0
			SSo6	35°51.6′ N	139°48.9′ E	2019	0	0	2	0	0	0	0
			SSo7	35°50.3′ N	139°49.6′ E	2019	0	0	1	0	0	0	0
			SSo8	35°50.7′ N	139°49.5′ E	2019	0	0	1	0	0	0	0
			SSo9	35°51.0′ N	139°49.3′ E	2019	0	0	3	0	0	0	0
			SSo10	35°50.8′ N	139°49.7′ E	2019	0	0	2	0	0	0	0
			SSo11	35°51.5′ N	139°50.1′ E	2019	0	0	1	0	0	0	0
			SSo12	35°51.6′ N	139°49.6′ E	2019	0	0	2	0	0	0	0
		Koshigaya	Sko1	35°52.3′ N	139°48.3′ E	2019	0	0	8	0	0	0	0
			Sko2	35°53.3′ N	139°48.0′ E	2019	0	0	1	0	0	0	0
Western Tokyo	Tokyo	Akiruno	TAk1	35°43.5′ N	139°19.5′ E	2017	4	0	0	0	0	0	0
						2018	5	0	0	0	0	0	0
		Fussa	TFu1	35°44.1′ N	139°19.2′ E	2019	4	0	0	0	0	0	0
Southern Tokai	Aichi	Tobishima	ATo1	35°04.9′ N	136°47.2′ E	2015	0	0	0	1	0	0	0
			ATo2	35°05.1′ N	136°46.9′ E	2017	0	0	0	1	0	0	0
			ATo3	NA	NA	2019	0	0	0	3	0	0	0
		Yatomi	AYa1	35°06.3′ N	136°44.9′ E	2019	0	0	0	5	0	0	0
			AYa2	35°04.8′ N	136°45.1′ E	2019	0	0	0	4	0	0	0
		Nagoya	ANa1	35°07.1′ N	136°52.3′ E	2019	0	0	0	3	0	0	0
Western Kansai	Osaka	Habikino	OHa1	34°32.1′ N	135°35.9′ E	2017	0	0	0	0	3	0	0
						2019	0	0	0	0	2	1	0
			OHa2	34°32.3′ N	135°35.5′ E	2019	0	0	0	0	1	2	0
Northern Tokushima	Tokushima	Itano	TIt1	NA	NA	2017	0	0	0	0	0	0	4

NA indicates that geographical information for sampling sites is not provided, in order to protect the privacy of the responsible landowners/organizations.

**Table 2 insects-13-00217-t002:** Nucleotide substitution sites in mitochondrial CO1 regions of *Aromia bungii* among the haplotypes detected in the present study. The shorter section was determined with the primer pair LCO1490 and HCO2198. The longer section was determined with the primer pair CO1-Corz and tRNA^Leu^-R.

Haplotype	Position in Shorter Section													Position in Longer Section						
	60	172	273	345	384	405	450	471	483	513	549	594	606	84	249	306	363	414	534	700
A	A	C	T	A	G	T	C	G	G	T	G	G	A	A	T	A	G	G	A	T
B	•	•	•	•	A	C	•	A	•	C	•	•	•	•	•	•	A	•	G	•
C	G	•	•	•	A	C	T	•	•	C	•	•	•	G	•	•	A	•	G	•
D	•	•	•	•	•	•	•	•	•	•	•	•	•	•	•	•	•	•	G	•
E	•	T	G	•	•	•	•	•	•	C	•	•	•	•	A	•	A	A	G	C
F	•	•	•	•	•	•	•	•	•	C	•	•	•	•	•	G	•	•	G	•
G	•	•	•	G	•	•	•	•	A	C	A	A	C	•	•	•	A	•	G	•

**Table 3 insects-13-00217-t003:** Molecular variance analysis results for the populations of *Aromia bungii* in northern Kanto and southern Saitama.

Source of Variation	df	Sum of Squares	Variance Component	Proportion of Variance	Φ
Between regions	1	531.9	14.0	72%	Φ_CT_ = 0.723 (*p* < 0.001)
Among sites within regions	24	151.0	0.5	3%	Φ_SC_ = 0.092 (*p* = 0.071)
Within sites	76	931.9	4.9	25%	

df, degrees of freedom; ΦCT, molecular variance among groups; ΦSC, molecular variance within groups.

## Data Availability

All data analyzed in this study are indicated in this article and the Appendix A, or have been deposited into the DNA Data Bank of Japan, European Molecular Biology Laboratory, and GenBank. GenBank accession numbers are indicated in-text and in Appendix A.

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
