# Peer review of "Genetic Differences among Established Populations of Aromia bungii (Faldermann, 1835) (Coleoptera: Cerambycidae) in Japan: Suggestion of Multiple Introductions"

_insects, 2022, doi:10.3390/insects13020217_

Round 1
Reviewer 1 Report
Review of Tamura S. and Shoda-Kagaya E: “Genetic differences among established populations of Aromia bungii (Coleoptera; Cerambycidae) in Japan: suggestion of multiple introductions”
This is an interesting study on the red-necked longhorn beetle (RLB) Aromia bungii, an invasive, wood-boring pest that has an increasing importance in the agriculture/forestry due to the increase of its distribution range. The data is interesting and worth publishing, and the aim of the work and potential significance of the results are clearly stated. The seven haplotypes revealed in the six regions of the country indeed seem to support the hypothesis of multiple introductions. The text is also correctly written in English.
However, there are some minor more general and detailed flaws that should be corrected and some issues that should be better explained.
The author should consider presenting a picture/plate with some of the studied individuals of A. bungii in the MS, which would also contribute to the higher interest of a reader; not everyone knows how this species looks like, which seems even more important in case of invasive species and may contribute to increased detectability of this species in the region.
Have you noticed any even the slightest differences in phenotype between the detected haplotypes? Size, colouration/shade, etc? Please add such a sentence to the text, regardless of the results.
Specific comments:
Line 3 and 36: Please provide full taxonomic authority: (Faldermann, 1835). Also, there should be a colon, not a semicolon, between “Coleoptera” and “Cerambycidae”.
Line 11: “last a decade” – it should be “last decade”.
Line 13: Please do not use the constructions as here: “A. bungii populations”. It always should be “populations of A. bungii”, “individuals of A. bungii”, etc. Please correct through the entire manuscript.
Lines 32/33: Keywords should be arranged in alphabetical order.
Line 56: Do the colours used for the haplotypes E, F, G suggest any greater similarity to the haplotypes A, B, C, respectively, due to similar colour used?
Line 67: The square brackets inside regular brackets look quite strange – perhaps “materials, e.g., [13–17]” or “[e.g., 13–17]” would be better.
Lines 88–92: There is no information on how the individuals were killed/preserved. Were they put directly to ethanol (what concentration?) in the field? Also, have the reared beetles been treated the same?
Line 93: Why did you decide to sequence exactly two sections of the mitochondrial DNA CO1 regions (612 and 762 bp)?
Line 127 (In Table 1. Sampling information and haplotype compositions 127 of Aromia bungii.): The total number of individuals collected in the South Saitama region (sites SSo1-1 to SKo2-1) (32) differs from that presented in the graph (30).
Lines 147–149: The first sentence needs to be transferred to the Materials and Methods.
Lines 150/151: There is a big mess with the tables numeration… First of all, both regular tables in the text are numbered as “Table 1” instead 1 & 2. Moreover, what is the difference between “Table A1” at the end of the MS and “Table S1” in Excel file? Both tables present different data so they should be clearly distinguishable. I believe that all supplementary materials (appendixes) should have an analogous and continued numbering.
Line 180: “were not consisted the sequences” – “not consisted OF the sequences”? Or rather “not consistENT WITH the sequences.
Line 213: Although the paper of Russo et al. 2020 was mentioned in the Discussion, it seems advisable to add some additional data from that paper and to confront the results of those authors with the results presented in the MS since both studies were very similar plus were conducted in very different regions of the world. In Europe, molecular analyses highlighted that all specimens recovered in Italy share the same haplotype (however different from the German one), which support that the invasive process in the country started from only one introduction. It would be interesting to emphasize this fact.

Reviewer 2 Report
- Line 67 - indicate in detail the individual bibliographic references: 13-17. Does it mean from number 13 to 17?
- Line 120 indicate the “g” or the centrifuge model (refrigerated or not?) Used
- Line 139: maybe countries?
- Indicate (if possible) differences in performance of the extraction methods used.
- Indicate whether a biophometer reading has been made for the quantification of the extracted DNA with relative qualitative data.
- Indicate the company and model of the centrifuges used.
- Indicate the company and model of the MilliQ distiller.
- Highlight, (if possible) also graphically the alignments produced with BioEdit ver. 7.2.5.
- Line 150: table A1? Maybe Table 1 column A?
